# Detection Model on Fatigue Driving Behaviors Based on the Operating Parameters of Freight Vehicles

**Jianfeng Xi** [1], **Shiqing Wang** [1], **Tongqiang Ding** [1,*], **Jian Tian** [2], **Hui Shao** [1] and **Xinning Miao** [1]

1  College of Transportation, Jilin University, Changchun 130022, China; xijf@jlu.edu.cn (J.X.);
   shiqing18@jlu.edu.cn (S.W.); shaohui18@jlu.edu.cn (H.S.); xnmiao20@jlu.edu.cn (X.M.)
2  China Academy of Transportation Sciences, Beijing 100029, China; tianjian@motcats.ac.cn
*  Correspondence: dingtq@jlu.edu.cn

**Abstract:** Whether in developing or developed countries, traffic accidents caused by freight vehicles are responsible for more than 10% of deaths of all traffic accidents. Fatigue driving is one of the main causes of freight vehicle accidents. Existing fatigue driving studies mostly use vehicle operating data from experiments or simulation data, exposing certain drawbacks in the validity and reliability of the models used. This study collected a large quantity of real driving data to extract sample data under different fatigue degrees. The parameters of vehicle operating data were selected based on significant driver fatigue degrees. The *k*-nearest neighbor algorithm was used to establish the detection model of fatigue driving behaviors, taking into account influence of the number of training samples and other parameters in the accuracy of fatigue driving behavior detection. With the collected operating data of 50 freight vehicles in the past month, the fatigue driving behavior detection models based on the *k*-nearest neighbor algorithm and the commonly used BP neural network proposed in this paper were tested, respectively. The analysis results showed that the accuracy of both models are 75.9%, but the fatigue driving detection model based on the *k*-nearest neighbor algorithm is more reliable.

**Keywords:** freight vehicles; fatigue driving behavior detection; *k*-nearest neighbor algorithm; BP neural network

## 1. Introduction

According to the National Highway Traffic Safety Administration [1] report, 11.2% of road traffic accidents in the United States in 2015 involved at least one large freight vehicle. In developing countries such as India, the situation is even worse. The National Crime Bureau of India (NCRB) [2] records that in 2015, commercial freight vehicles accounted for 19.4% of the total fatalities in traffic accidents. At the same time, with the acceleration of globalization, the demand for road freight transportation will further increase, and drivers often sacrifice sleep and rest time to drive for a long time under the pressure of life. Fatigue driving is one of the main causes of traffic accidents of freight vehicles [3].

The fatigue state of freight vehicle drivers has unique characteristics due to their occupational characteristics. In 2013, the US Centers for Disease Control and Prevention [4] stated that commercial freight drivers were more likely to be drowsy or fatigued than other drivers. Through questionnaires, Feyer, A.M. [5] found that the driving experience of long-distance freight drivers and long-distance bus drivers only partially overlaid. Compared with bus drivers, most freight drivers said that they often felt fatigue, while relatively few bus drivers said that fatigue was the main problem of their driving. Fitzharris, M. et al. [6] found that compared to not providing feedback to drivers when fatigue was detected, providing feedback to the company and the cockpit in real time could effectively reduce the occurrence of fatigue driving events. In summary, it is of great significance to quickly and accurately detect the fatigue driving behavior of freight vehicles.

At present, the research on freight vehicle fatigue driving mainly focuses on the correlation between driver fatigue and macro factors such as daily work and rest, work

pressure, management system, physical quality, etc. [7–9]. Adams, G.J. et al. [10] studied the correlation between freight driver fatigue, work organization, accident experience, etc. The results showed that driver fatigue was significantly correlated with rest, transportation route arrangement, and management system. Wiegand, D.M. et al. [11] invited professional freight drivers to participate in a long-term natural driving study to explore the relationship between the body mass index of commercial freight drivers and fatigue driving events, and the results showed that obese individuals were more likely to be fatigued. However, there are few studies on the detection of fatigue degrees of freight vehicle drivers. Wang, Y. et al. [12] carried out 2 h, 3 h, and 4 h natural driving tests of commercial drivers to test the correlation between drivers' visual behaviors and subjective fatigue degree.

According to sensor data source of the fatigue detection system, the existing fatigue driving behavior detection systems are mainly divided into two types: invasive and non-invasive [13,14]. The invasive fatigue detection system collects physiological data such as electroencephalogram (EEG), electrocardiogram (ECG), electrooculogram (EOG), heart rate detection, and other physiological data of the driver; analyzes the physiological characteristics of the driving process; and realizes the detection of the driver's fatigue degrees [15–17]. However, the intrusive fatigue driving detection requires contact with the driver's body, which may easily lead to distraction or discomfort for the driver; therefore, its application in real-life situations is restricted. The non-intrusive systems primarily detect the driver's fatigue degrees by extracting the driver's facial features, vehicle operating data, and other information. During the detection process, the driver's normal driving will not be disturbed, and the data acquisition and fatigue detection can objectively reflect the driver's actual operating behavior or states [18,19]. Therefore, at present, the non-intrusive methods have become more popular in the field of driver fatigue detection, and most of them obtain data through simulated driving or real vehicle experiments.

From the perspective of fatigue driving detection in a simulated driving environment, Ji, Y. et al. [20] proposed a detection algorithm based on multi-index fusion and state recognition network. In this study, a multi-task level convolutional neural network was used to detect faces and feature points, and a fatigue detection model was established by combining the two features of eyes and mouth states. Wang, M. et al. [21] input different combinations of vehicle operating parameters into the fatigue driving detection model based on the random forest algorithm, and the research results show that the fatigue degrees of drivers can be detected according to the three indexes of steering angle, lateral acceleration, and longitudinal acceleration. McDonald, A.D. et al. [22] took the time dependence of the transition between sleepy and awake states into consideration, and designed the dynamic Bayesian network fatigue detection algorithm with the steering angle, pedal, vehicle speed, and acceleration as inputs, which improved the detection accuracy. From the perspective of experimental fatigue driving detection of real vehicles, Li, Z. et al. [23] detected the fatigue degree of drivers based on different steering angles and yaw angles in the real driving environment, and the detection accuracy reached 88.02%. Al-Libawy, H. et al. [24] assuming that vehicle acceleration, vehicle rotation mode, driver's head position, and head rotation are fatigue related indicators, and developed a fatigue driving detection system. The above research results show that the fatigue degree of the driver can be detected by using the parameters of the vehicle such as steering angle, lateral position, heading angle, and acceleration, and the detection accuracy of the model will be improved based on multiple parameters. However, most of the above research data are from driving simulator environments or real vehicle experiments. On the one hand, the reliability of a fatigue monitoring system developed on this basis still lacks effective verification under real vehicle conditions, and on the other hand, the number of subjects is small.

From the point of the principle of the detection method, the vast majority of current fatigue driving detection studies have used the machine learning and deep learning algorithms, including cascade regression tree, decision tree, neural network, support vector machine (SVM), random forests, Bayesian networks, etc., and combine the collected data

of face, physiological, and vehicle operation to detect fatigue driving [25–28]. However, first of all, it is difficult to calibrate some parameters accurately. For example, in the sample training process, some basic parameters and training function parameters are randomly generated, the training efficiency is low, and the influence of the number of training samples and other parameters on the model detection accuracy and model stability is seldom considered. Secondly, the intermediate learning process cannot be observed and the output results are difficult to explain, which will affect the credibility and acceptability of the results. Finally, the requirements for data quality and content are relatively high, and the practical application is more difficult.

With the extensive application of various vehicle operation management systems, vehicle operation data and drivers' facial monitoring video on the actual road can be acquired in real time and stored for a long time, which greatly improves the richness and objectivity of data sources [29]. At the same time, the *k*-nearest neighbor algorithm is the most intuitive in principle and the method is simple. It depends on the number of surrounding training samples, and the established model has strong stability [30]. Therefore, this paper collects a large number of continuous operating data of freight vehicles and driver video data under real road traffic conditions, extracts fatigue driving detection parameters from vehicle operating data, and studies a more accurate and stable fatigue driving detection model by using the *k*-nearest neighbor algorithm.

The paper is organized as follows. Section 2 extracts driver fatigue degree data, preprocesses the vehicle operating data, extracts the vehicle operating parameters that are significant to the fatigue degrees, and constructs the fatigue driving detection model. Section 3 compares and evaluates the performance of the model established in this paper by substituting data into the two models. Section 4 presents the conclusion.

## 2. Materials and Methods

### 2.1. Extraction of Driver Fatigue Degrees Data

The expert scoring method based on facial video is the most practical method for evaluating the fatigue degrees of drivers. This method uses a group of trained experts to score the fatigue degrees of drivers according to their facial expressions and head posture [31]. In this paper, the evaluation standard and process of driver fatigue degree is formulated as shown in Table 1.

**Table 1.** Evaluation criteria of driver fatigue degree.

| Fatigue Degree | Scoring | Character Description |
|---|---|---|
| Alert | 1 | Eyes are opened normally, blinking quickly, eyeballs are active, keeping attention to the outside world, head is straight, facial expressions natural. |
| Fatigue | 2 | Eyes tend to close, blinking slower, eye activity decreases, gaze is sluggish, yawning, deep breathing, sighing, swallowing, winking, shaking head, scratching, paying less attention to the surrounding environment. |
| Severe fatigue | 3 | Eyes are closed seriously and cannot be opened, eyes closed for a long time, nodding in a nap, head tilted, loss of ability to continue driving. |

The process of determining the driver fatigue degree: the duration of the collected driver's facial video is 20 s, which is called the sample. The samples are scored by three experts according to the evaluation criteria of the driver fatigue degree. If the three experts score the same, then the experts' consensus evaluation result is taken as the fatigue level of the sample. When the scoring results are inconsistent, the three experts will conduct a collegiate discussion. If they are consistent after the negotiation, the result will be used as the fatigue degree of the sample. If there is still a disagreement after the negotiation, the sample will be discarded.

*2.2. Extraction of Vehicle Operating Parameters*

2.2.1. Data Preprocessing

The data sampling frequency of the vehicle operating parameters is 1 s, and the driving duration is calculated according to the vehicle operating time and speed value. Taking into account factors such as data collection errors, short-term temporary parking, and other factors that do not relieve driver fatigue, when the duration operating velocity is continuously 0 km/h and does not exceed 200 s, then the vehicle is regarded as in a temporary stop state. This paper considers that the vehicle is still in operation. If the calculated driving duration for some samples is less than 300 s, it is still regarded as the vehicle is not running. In addition, if the driving duration of some samples is longer than 300 s, but the speed at each moment during the driving duration is lower than 10 km/h, it is still regarded as the vehicle is not running. Due to the large amount of data, the Python software is used to write a program to calculate the driving duration and extract the start and end time of each segment of driving.

The operating parameters of the vehicle directly obtained from the system mainly include heading angle, roll angle, speed, lateral acceleration, and longitudinal acceleration. By deriving the heading angle, roll angle, and speed, the heading angular velocity, roll angular velocity, and acceleration can be obtained. In this way, eight operating parameters of the vehicle can be obtained. As the degree of fatigue deepens, the driver's ability to perceive, react, and judge will become weaker, resulting in abnormal fluctuations in vehicle control variables and state variables. In this paper, by monitoring the heading angle, roll angle, speed, lateral acceleration, and longitudinal acceleration of the vehicle, the heading angle, roll angle, and velocity parameter data are derived to obtain the heading angular velocity, roll angular velocity, and acceleration, and the vehicle is monitored through these parameters: whether there is a deviation in the trajectory of the vehicle, whether the vehicle is driving smoothly, and whether there are behaviors such as sudden braking, rapid acceleration, sharp turning, and rapid lane change. Taking the heading angle $\alpha$ by deriving to obtain the heading angular velocity $\beta$ as an example, since the vehicle operating parameter is a smooth discrete function and the information density is large enough (the time interval is 1 s), the increment of the independent variable when deriving the derivation is $\Delta t = 1$, and the specific calculation is Formula (1) [32].

Take the vehicle heading angle $\alpha$ as an example:

$$\beta = \frac{d\alpha}{dt} = \lim_{\Delta t = 1} \frac{\alpha_t - \alpha_{t-1}}{\Delta t} = \frac{\alpha_t - \alpha_{t-1}}{1} = \alpha_t - \alpha_{t-1} \tag{1}$$

where $\beta$ is the current heading angular velocity of the vehicle at the current moment $t$; $\alpha_t$ is the current heading angle value at moment $t$; and $\alpha_{t-1}$ is the vehicle heading angle value at moment $t - 1$.

On the basis of obtaining the eight operating parameters of the vehicle, the absolute mean value and standard deviation of each parameter are calculated under different time windows (5 s, 10 s, 15 s, 20 s, and 30 s), so that a total of eighty operating parameters of the vehicle are obtained. Taking the heading angle $\alpha$ as an example, the calculation method is shown in Formulas (2) and (3).

$$\alpha_{mean} = \frac{1}{N} \sum_{i=1}^{N} |\alpha_i| \tag{2}$$

where $\alpha_{mean}$ is the absolute mean value of heading angle; $N$ is the length of the time window, with the $N$ values of 5, 10, 15, 20, and 30; and $\alpha_i$ is the heading angle value of the $i$-th sample.

$$\alpha_{STD} = \sqrt{\frac{1}{N-1}(\alpha_i - \alpha_m)^2} \tag{3}$$

where $\alpha_{STD}$ is the standard deviation of heading angle; and $\alpha_m$ is the mean value of the heading angles.

$$\alpha_m = \frac{1}{N}\sum_{i=1}^{N}\alpha_i \tag{4}$$

### 2.2.2. Parameter Extraction

Firstly, the normal distribution test and the variance homogeneity test are performed on the vehicle operating parameters under different fatigue degrees. When these two conditions are met, the single-factor variance method is used to study the difference of vehicle operating parameters under different fatigue degrees. When these two conditions are not met, the Friedman test is used instead [33,34]. Then, the Bonferroni-adjusted multiple comparison test analysis method is used to compare the differences of each parameter under the three different fatigue degrees. The significance level of the one-way analysis of variance and the Friedman test in this study is 0.05, and the significance level of the Bonferroni-adjusted multiple comparison test is 0.01. The test results show that fatigue status has a significant impact on the following nine vehicle operating parameters, $p < 0.001$ (See Table 2).

**Table 2.** Statistical analysis results of vehicle operating parameters (significantly affected) under different fatigue degrees.

| Vehicle Operating Parameters | Time Window (S) | Mean (Standard Deviation) | | | $\chi^2$/F | Multiple Comparison Test (Z Statistics) | | |
| --- | --- | --- | --- | --- | --- | --- | --- | --- |
| | | Severe Fatigue | Fatigue | Alert | | Fatigue-Severe Fatigue | Alert-Severe Fatigue | Alert-Fatigue |
| Standard deviation of heading angle | 20 | 4.621 (3.946) | 1.655 (1.482) | 0.465 (0.421) | 167.227 | −6.093 | −12.925 | −6.832 |
| Absolute mean value of heading angular velocity | 20 | 0.951 (1.117) | 0.488 (1.238) | 0.161 (0.091) | 180.424 | −6.155 | −13.417 | −7.262 |
| Standard deviation of heading angular velocity | 20 | 1.482 (4.026) | 0.981 (5.201) | 0.240 (0.130) | 169.561 | −5.662 | −12.986 | −7.324 |
| Standard deviation of Roll angle | 20 | 0.866 (0.465) | 0.649 (0.274) | 0.462 (0.233) | 63.015 | −2.708 | −7.861 | −5.108 |
| Absolute mean value of roll angular velocity | 20 | 0.467 (0.182) | 0.374 (0.132) | 0.312 (0.131) | 47.591 | −4.247 | −6.832 | −3.585 |
| Absolute mean value of lateral acceleration | 30 | 0.644 (0.106) | 0.538 (0.081) | 0.434 (0.061) | 161.106 | −5.108 | −12.617 | −7.509 |
| Standard deviation of lateral acceleration | 15 | 0.616 (0.137) | 0.463 (0.100) | 0.355 (0.083) | 167.455 | −5.908 | −12.925 | −7.016 |
| Standard deviation of roll angular velocity | 15 | 0.588 (0.287) | 0.468 (0.192) | 0.383 (0.187) | 35.318 | −3.400 | −5.908 | −3.508 |
| Driving duration | / | 4.033 (1.139) | 3.879 (2.590) | 1.949 (0.794) | 146.645 | −6.532 | −11.417 | −8.893 |

### 2.3. Construction of Detection Model on Fatigue Driving Behaviors

#### 2.3.1. Model Construction

The *k*-nearest neighbor algorithm is used to construct the fatigue driving detection model. The core idea of the algorithm is that if the fatigue degree of the sample isunknown and most *k*-nearest samples of the sample in the feature space belong to a certain fatigue degree, then the sample also belongs to this degree. The *k* value is usually an integer not greater than 20 [35]. In order to enhance the accuracy and reliability of the *k*-nearest neighbor algorithm, the work considers the influence of the *k* value and the number of training samples on the accuracy of fatigue detection.

The extracted sample set *X* of different fatigue degrees constitutes a matrix of $n \times (m+1)$ (see Formula (5)). Since each vehicle operating parameter represents different attributes, and the value of each parameter has a different order of magnitude and value range, it is necessary to normalize each column of the parameter values before using the algorithm. In

this paper, the maximum-minimum method is used for normalization (see Formula (6)), and the normalized sample set $X^*$ of three fatigue degrees is obtained [36].

$$
X = \begin{bmatrix}
x_{11} & \cdots & x_{1t} & \cdots & x_{1m} & x_{1m+1} \\
 & & \vdots & \vdots & \vdots & \\
x_{k1} & \cdots & x_{kt} & \cdots & x_{km} & x_{km+1} \\
 & & \vdots & \vdots & \vdots & \\
x_{n1} & \cdots & x_{nt} & \cdots & x_{nm} & x_{nm+1}
\end{bmatrix}
\tag{5}
$$

where $X$ is the sample set; $x_{kt}$ is the $t-$th parameter value in the $k-$th sample ($k = 1, 2, \cdots, n$; $t = 1, 2, \cdots, m$); and $x_{km+1}$ is the fatigue degree of the $k-$th sample.

$$
x_{kt}^* = \frac{x_{kt} - x_{t(min)}}{x_{t(max)} - x_{t(min)}}
\tag{6}
$$

where $x_{kt}^*$ is the $t-$th parameter value in the $k-$th sample in the sample set $X^*$; $x_{t(min)}$ represents the minimum value of the $t-$th parameter in the sample set $X$; and $x_{t(max)}$ represents the maximum value of the $t-$th parameter in the sample set $X$.

The improved $k$-nearest neighbor algorithm is solved as follows:

1.  Extract sample training set and sample test set.

    Firstly, the first $l$ samples are extracted from the sample set $X^*$ to constitute a test sample set $P$ whose fatigue degree is unknown. Each test sample is denoted as $x_r^* = \{x_{r1}^* \cdots x_{rt}^* \cdots x_{rm}^* x_{rm+1}^*\}$. Then, the $q$ samples are extracted from the rest samples as the training sample set Q with known fatigue degree, where $q < n - l$. Each training sample is denoted as $x_{l+s}^* = \{x_{l+s\,1}^* \cdots x_{l+st}^* \cdots x_{l+sm}^* x_{l+sm+1}^*\}$. Finally the $k$ value in the $k$-nearest neighbor algorithm is determined.

2.  For each test sample $x_r^*$ whose fatigue degree is unknown, perform the following operations in sequence:

    a.  Calculate the distance between each sample $x_{l+s}^*$ in the training set and the test sample $x_r^*$, using the Euclidean distance calculation (see Formula (7)).

$$
d_{rl+s} = \sqrt{\sum_{t=1}^{m} \left( x_{rt}^* - x_{l+st}^* \right)^2}
\tag{7}
$$

    where $d_{rl+s}$ is the Euclidean distance between the test sample $x_r^*$ and the training sample $x_{l+s}^*$; $r$ is $1, 2, \cdots, l$; $l + s$ is $l + 1, l + 2, \cdots, n - l$; $m$ is the number of parameters in each sample; $x_{rt}^*$ is the $t-$th parameter value of the test sample $x_r^*$; and $x_{l+st}^*$ is the $t-$th parameter value of the training sample $x_{l+s}^*$.

    b.  Sort the distances calculated by each training sample and test samples in ascending order.

    c.  Select the first $k$ training samples with the smallest distance from the test samples.

    d.  Determine the frequency of occurrence of each fatigue degree in the first $k$ training samples.

    e.  Take the fatigue degree with the highest frequency in the first $k$ samples as the fatigue degree of the test sample.

3.  Compare whether the actual fatigue degree of all the test samples is consistent with the predicted fatigue degree. Then calculate the number of correct predictions of various samples in the test sample.

4.  Adjust the number $q$ of training samples, and then perform steps (1), (2), and (3). The cycle ends when all the different training samples numbers ($q$ value) are executed. The $q$ value range is $0 - (n - l)$.

5.  Change the $k$ value and then perform steps (1), (2), (3), and (4). The cycle ends when all the different $k$ values (1, 3, 5, 7, 9, 11, 13, 15, 17 and 19) are executed.

2.3.2. Model Performance Evaluation

In order to evaluate the performance of the model, firstly, according to the prediction results, the results of various fatigue degrees are calculated, and the confusion matrix (see Table 3) is listed. Then, the *Precision, True Positive Rate (TPR), and Truth Negativity Rate (TNR)* of each fatigue degree are calculated, respectively (see Formulas (12)–(14)).

$$Precision = \frac{TP}{TP + FP} \times 100\% \tag{8}$$

$$Sensitivity(TPR) = \frac{TP}{TP + FN} \times 100\% \tag{9}$$

$$Specificity(TNR) = \frac{TN}{TN + FP} \times 100\% \tag{10}$$

where $TP$ is the number of correctly predicted samples that actually belong to the fatigue degree. $FP$ is the number of samples that do not actually belong to the fatigue degree but are incorrectly predicted to the degree. $FN$ is the number of samples that belong to the fatigue degree and are mispredicted to other degrees. $TN$ is the number of correctly predicted samples that do not belong to the fatigue degree.

**Table 3.** Confusion matrix.

|  |  | Prediction | | |
|---|---|---|---|---|
|  |  | **Alert** | **Fatigue** | **Severe Fatigue** |
|  | Alert | a | b | c |
| Actual | Fatigue | d | e | f |
|  | Severe fatigue | g | h | i |

The BP neural network model can also effectively detect the fatigue driving behavior. Therefore, in order to test the effectiveness, the fatigue driving behavior detection model based on the *k*-nearest neighbor algorithm can be compared with the existing fatigue driving behavior detection model based on the BP neural network to test the effectiveness of the proposed method.

## 3. Results and Discussion

### 3.1. Data Source

The research data were derived from vehicle operating data and drivers' facial videos of 50 freight vehicles provided by the vehicle cloud control platform of a logistics company for nearly one month. The driving path is an expressway in Shandong Province. Take the Yanhai Expressway as an example, the driving path is shown in the Figure 1. Through the evaluation of the drivers' fatigue degree and the extraction of significant parameters of vehicle operating data, a total of 396 samples data were obtained, of which the number of samples for each fatigue degree was 132. Each data sample contains nine operating parameters of vehicles that are significantly affected by the fatigue degrees and one driver fatigue degree.

For the convenience of the research, the order of 396 samples in the sample set X* was randomly disrupted. The first 79 samples (20% of the total sample number) were taken as the model test sample set, that is, the value of variable L in Formula 7 is 79.Therewere 24 alert samples, 24 fatigue samples, and 31 severe fatigue samples. The remaining samples served as the training sample set.

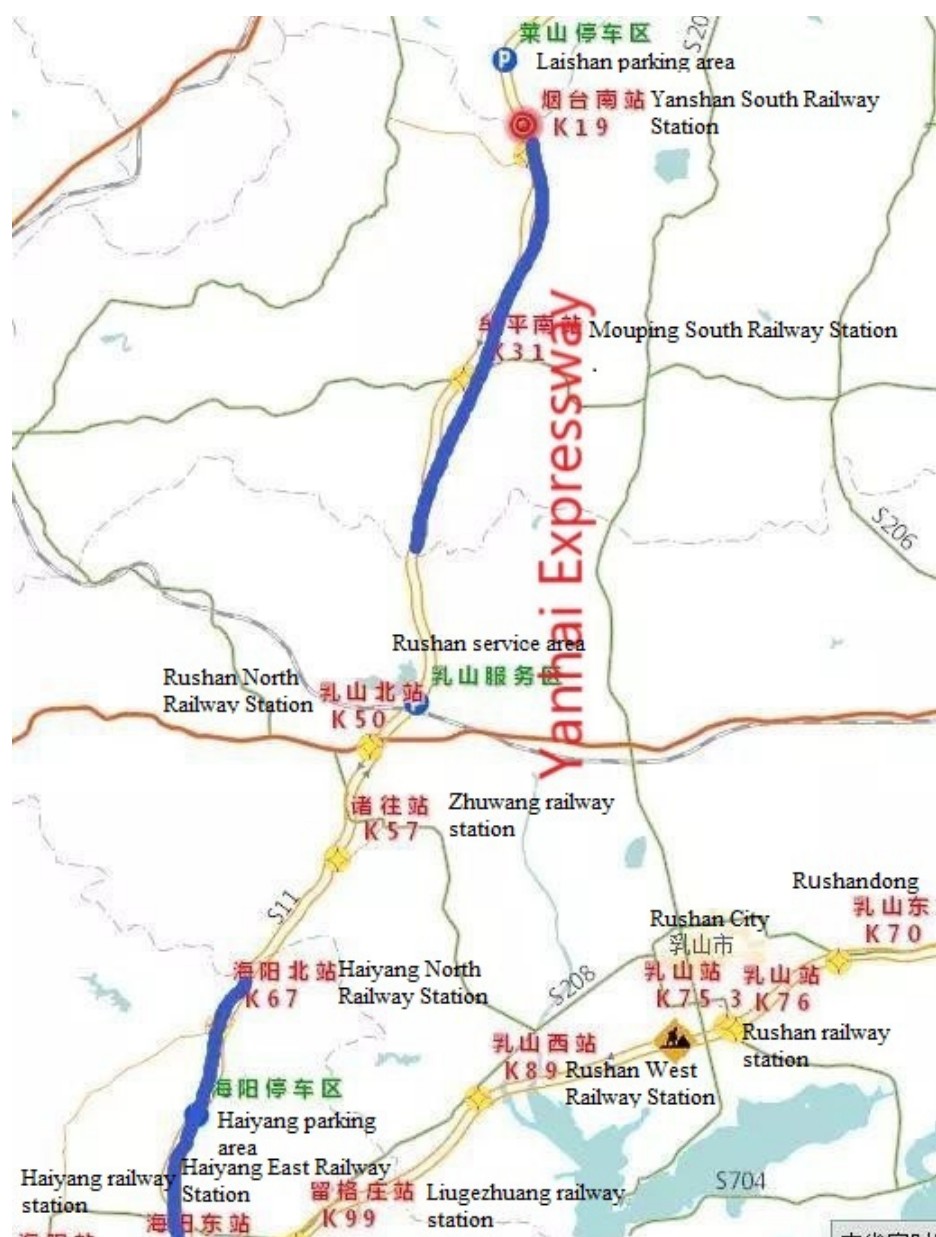

**Figure 1.** Vehicles' driving path.

### 3.2. Discussion of Experimental Results

In order to test the effectiveness of the fatigue driving behavior detection model based on the *k*-nearest neighbor algorithm, the model (referred to as Model 1) was compared with the existing fatigue driving behavior detection model based on the BP neural network (referred to as Model 2). Both models considered the influence of the number of different training samples and other parameters on the detection accuracy.

1.  Analysis of the experimental results Model 1

The program is developed with Python to realize the calculation of the fatigue driving detection model based on the *k*-nearest neighbor algorithm. According to the data obtained above, the parameters of the model can be known: the number of test sample $l = 79$, with the number of training sample $q \leq 317$, and the number of vehicle operating parameters significantly affected by fatigue degree $m = 9$. The value of $k$ is usually not greater than 20, and the values are 1, 3, 5, 7, 9, 11, 13, 15, 17 and 19, respectively. The accuracy of

fatigue driving behavior detection that changes with the number of training samples is calculated and shown in Figure 2. The number range of training samples corresponding to the maximum detection accuracy is shown in Table 4.

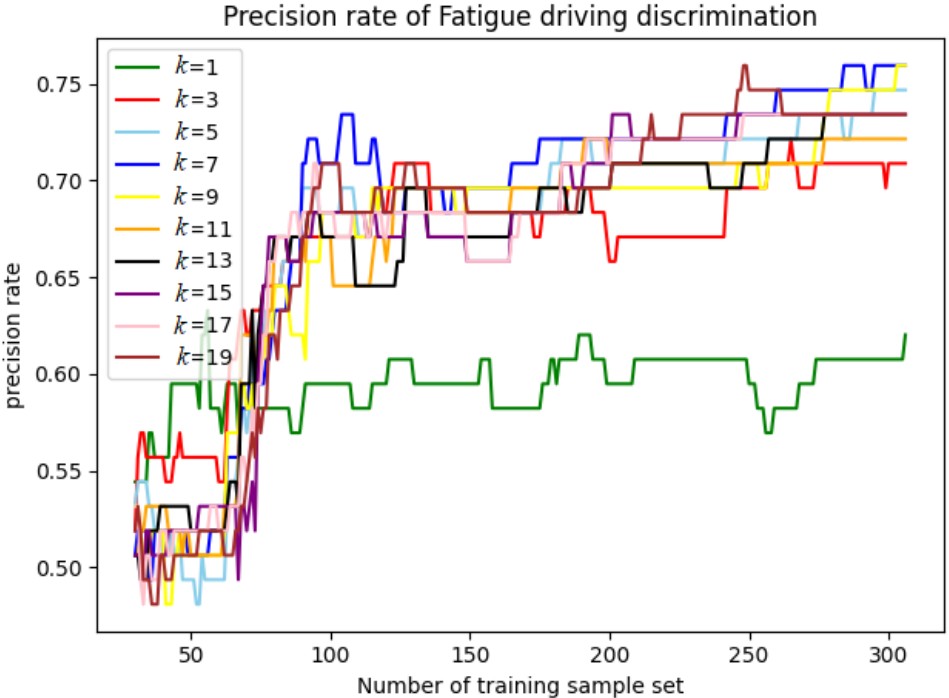

**Figure 2.** Accuracy of fatigue driving behavior detection with the number of training sample when $k$ takes different values.

**Table 4.** The maximum value of the fatigue driving detection accuracy when $k$ takes different values and the corresponding number of training samples.

| $k$ Value | Number of Training Sample Corresponding to the Maximum Detection Accuracy | Detection Accuracy |
|---|---|---|
| 1 | 56 | 63.3% |
| 3 | 265 | 72.3% |
| 5 | 295–306 | 74.7% |
| 7 | 284–306 | 75.9% |
| 9 | 303–306 | 75.9% |
| 11 | 191–199, 277–306 | 72.2% |
| 13 | 277–306 | 73.4% |
| 15 | 246–306 | 73.4% |
| 17 | 248–306 | 73.4% |
| 19 | 248–249 | 75.9% |

Figure 2 and Table 4 show that when $k$ values (smaller values) are 1 and 3, the detection accuracy fluctuates greatly with the change of the number of training samples, and the accuracy is relatively low. When $k$ values (larger values) are 5, 7, 9, 11, 13, 15, 17, and 19, and the number of the training samples isgreater than 70, the detection accuracy improves as the number of training samples increase and, until the number of training sample is greater than 200, the accuracy reaches more than 70.0% with a stable trend. When the $k$ value is 7 with the number range of training samples of 284–306, the accuracy of fatigue driving detection reaches the maximum value of 75.9%. Therefore, the $k$ value of the model should be set to 7, and when the ratio of the test set to the training set is between (2:7)–(2:8), the accuracy is high with strong stability so that requirements for training samples are low.

When the detection accuracy reaches the maximum, the detection results of each fatigue degree are shown in Table 5. By calculating the *precision, true rate (TPR), and true*

*negative rate (TNR)* of each type of fatigue degree prediction, the algorithm is evaluated (see Table 6). As can be seen from Table 6, when the driver is alert, the three index values of the evaluation algorithm are all greater than or equal to 87.5%, indicating that the algorithm can distinguish the driver's alert state and non-alert state well. When the driver is in the state of fatigue and severe fatigue, the *TNR* is larger, while the *Precision* and *TPR* are smaller, indicating that the algorithm has a higher probability of correctly identifying the state of non-fatigue or non-severe fatigue, but a lower probability of correctly identifying the state of certain fatigue or severe fatigue. In addition, combined with Table 5, it can be seen that the fatigue state and severe fatigue state are easily confused in detection. There are two reasons for this result. First, based on the videos, the scoring method is used to evaluate the driver's fatigue state with subjective factors. Second, the vehicle operating data of the fatigue state and severe fatigue state have no significant difference. On the whole, the model test results are reasonable.

**Table 5.** Confusion matrix based on the *k*-nearest neighbor algorithm (the test results when the test accuracy reaches the maximum).

|  |  | Prediction | | |
| --- | --- | --- | --- | --- |
|  |  | **Alert** | **Fatigue** | **Severe Fatigue** |
| | Alert | 21 | 3 | 0 |
| Actual | Fatigue | 1 | 18 | 5 |
| | Severe fatigue | 2 | 8 | 21 |

**Table 6.** *k*-nearest neighbor algorithm evaluation (the test results when the test accuracy reaches the maximum).

|  | *Precision* | *TPR* | *TNR* |
| --- | --- | --- | --- |
| Alert | 87.5% | 87.5% | 92.9% |
| Fatigue | 62.1% | 75.0% | 79.2% |
| Severe fatigue | 80.7% | 67.7% | 88.6% |

2. Analysis of the experimental results Model 2

The program is developed with Matlab to realize the calculation of the fatigue driving detection model based on the BP neural network (see Figure 3). From the data obtained above, the parameters of the model can be known: the number of the test sample $l = 79$, with the number of training sample $q \leq 317$. The number of input layer nodes (vehicle operating parameters), the number of output nodes (fatigue degree), the number of hidden layer nodes, and the learning efficiency are set as 9, 3, *midnum*, and 0.1, respectively. The neural network structure of $9 - midnum - 3$ is adopted, and the training times are set as 10, 50, 100, 200, and 500, respectively [37]. The random number between $(-1,1)$ is selected as the initial value of connection weight among the neurons of the input layer, hidden layer, and output layer, as well as the initial value of the threshold values of the hidden layer and output layer [38,39]. The Sigmoid activation function is used as the excitation function of the hidden-layer neurons. Referring to the constraints of the number setting of the hidden-layer nodes in the BP neural network algorithm, it is determined that the maximum and minimum numbers of nodes were 7 and 3, respectively. Therefore, the number range of the hidden-layer nodes is 3–7 [40].

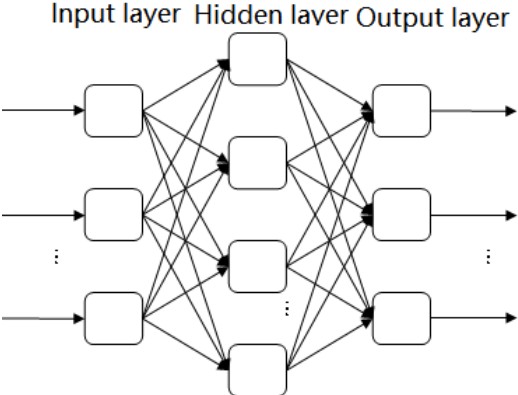

**Figure 3.** BP neural network structure.

Under the premise of setting the training times as 10, 50, 100, 200 and 500, respectively, the different numbers of hidden layer nodes are taken to obtain the accuracy of fatigue driving detection with the different ratios of training sample and test sample (see Figures 4–8). No matter what the value of the training time is, when the detection accuracy reaches the maximum, the number of hidden layer nodes is 7. Therefore, the number of hidden layer nodes is determined to be 7. In addition, the model requires higher training data. Although the ratios of the test sample to the training sample are at 4:6, 3:7, and 2:8, the detection accuracy is above 70%, yet among these ratios, the detection accuracy fluctuates greatly.

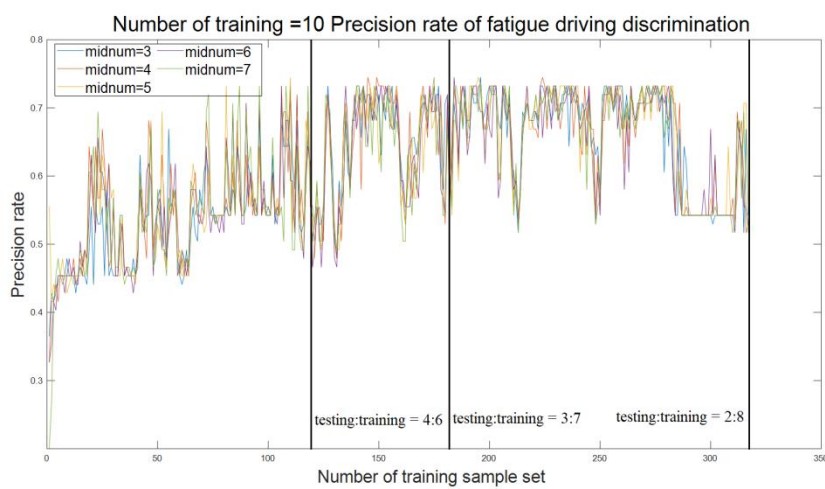

**Figure 4.** Accuracy of fatigue driving behavior detection varying with the number of training samples when the training times is 10.

When the training times (smaller values) are 10 or 50 (see Figures 4 and 5), the detection accuracy is 75.9%. However, the fluctuation is large, the ratio of training samples to detection samples is high, and the reliability of the model is poor. When the training times are 100 or 200 (see Figures 6 and 7), the detection accuracy fluctuates greatly at first, and then it will fluctuate between 60.0–75.9% with the number of training sample increasing. However, there are still some large sudden changes in the intermediate detection accuracy, resulting in poor reliability. When training times are 500 (see Figure 8), the accuracy is generally stable and fluctuates between 60.0–75.9%. Reliability is generally acceptable, but the number of training samples corresponding to the maximum detection accuracy is still small. In addition, the model needs to consider a lot of parameters. Some parameters are randomly generated, leading to one operating result inconsistent with the previous result. The model structure is complex, which takes a long operating time.

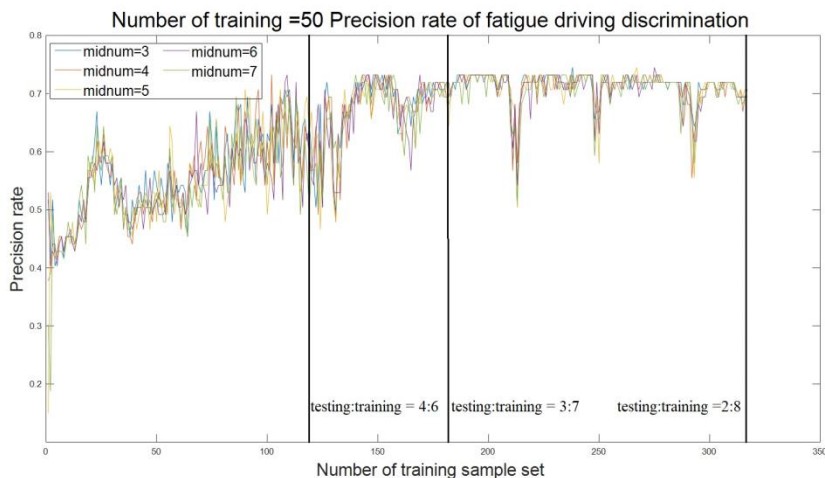

**Figure 5.** Accuracy of fatigue driving behavior detection varying with the number of training samples when the training times is 50.

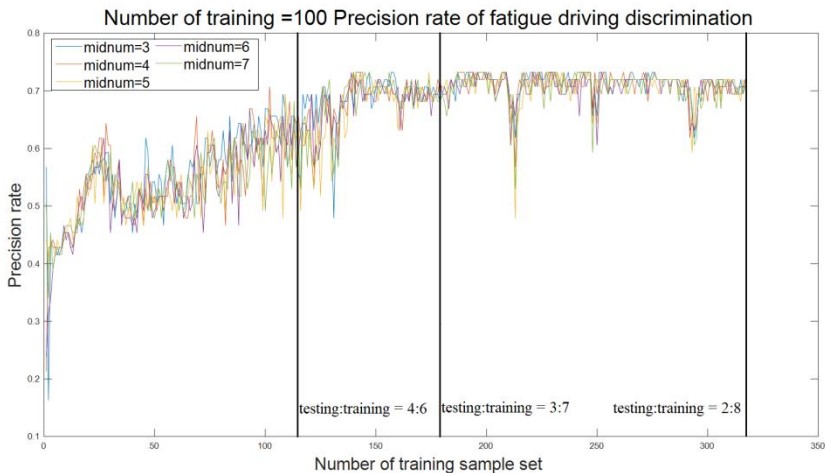

**Figure 6.** Accuracy of fatigue driving behavior detection varying with the number of training samples when the training times is 100.

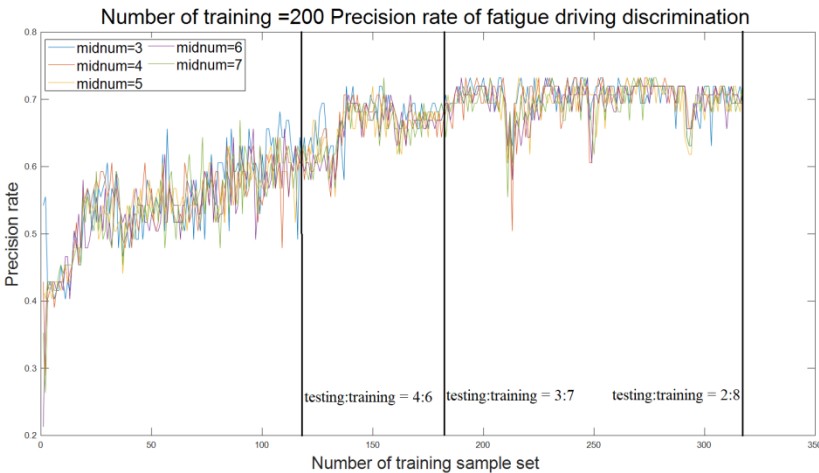

**Figure 7.** Accuracy of fatigue driving behavior detection varying with the number of training samples when the training times is 200.

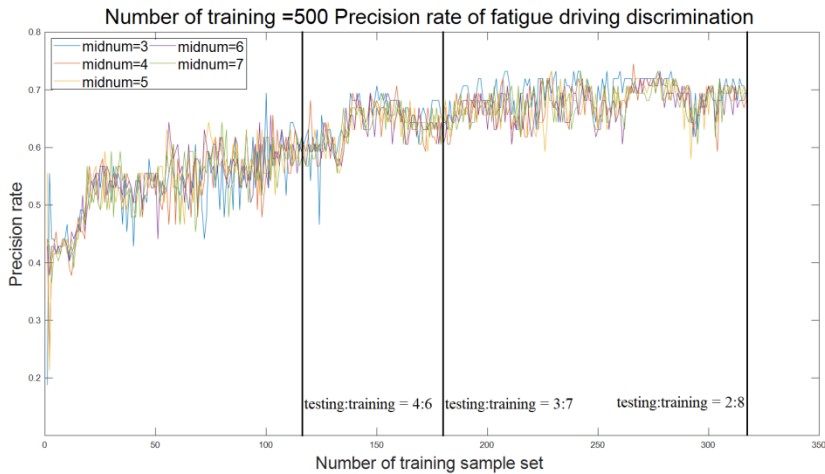

**Figure 8.** Accuracy of fatigue driving behavior detection varying with the number of training samples when the training times is 500.

In order to reflect the overall effect of the model, under the premise that the training times are set as 10, 50, 100, 200, and 500, respectively, and the number of hidden nodes is 7, the mean value of detection accuracy under the numbers of different training sample is calculated (see Table 7). It can be seen from Table 7 that when the training times are 50, the mean value of the detection accuracy is very small, less than 60%. When the training times are other values, the average detection accuracy is relatively large, about 64%, but the overall stability of the model is poor.

**Table 7.** The mean value of detection accuracy.

| Training Times | 10 | 50 | 100 | 200 | 500 |
|---|---|---|---|---|---|
| the mean value of detection accuracy | 57.4% | 64.8% | 64.5% | 63.7% | 62.3% |

When the detection accuracy reaches the maximum, the detection results of each fatigue degree are shown in Table 8. By calculating the *Precision*, *TPR*, and *TNR* of each type of fatigue degree prediction, the algorithm is evaluated (see Table 9). Comparing Table 9 with Table 6, it can be seen that there is little difference in the detection results of the two models. When the driver is in the alert state, the evaluation index value of the algorithm is relatively large, which can correctly distinguish the alert state and the non-alert state. In the fatigue state and the severe fatigue state, the *TNR* is larger, while the *Precision* and the *TNR* are smaller, indicating that the algorithm has a higher probability of correctly identifying the state of non-fatigue or non-severe fatigue, but a lower probability of correctly identifying the state of certain fatigue or severe fatigue.

**Table 8.** Confusion matrix based on the BP neural network (the test results when the test accuracy reaches the maximum).

| | | Prediction | | |
|---|---|---|---|---|
| | | **Alert** | **Fatigue** | **Severe Fatigue** |
| | Alert | 23 | 1 | 0 |
| Actual | Fatigue | 2 | 17 | 5 |
| | Severe fatigue | 2 | 8 | 21 |

**Table 9.** BP neural network algorithm evaluation (the test results when the test accuracy reaches the maximum).

|  | *Precision* | *TPR* | *TNR* |
|---|---|---|---|
| Alert | 85.1% | 95.8% | 90.5% |
| Fatigue | 65.4% | 70.8% | 83.0% |
| Severe fatigue | 80.8% | 67.7% | 88.9% |

3. Comparative analysis of two models

The above results show that the accuracy of the two models could reach 75.9% by taking different parameter values and different numbers of training samples. The fatigue driving detection model based on the *k*-nearest neighbor algorithm has reasonable detection results. With the increase of the training sample number, the accuracy of fatigue driving detection increases regularly, and the number of training samples corresponding to the maximum detection accuracy isin a certain interval. Therefore, the detection model based on the *k*-nearest neighbor algorithm is simple in principle and reliable. The fatigue driving behavior detection model based on the BP neural network has a complicated structure, which takes a long time to detect. The parameters are randomly selected, and the model requires higher training data, leading to the great fluctuation of detection results with sufficient reliability. The maximum accuracy values of the two models are the same, and the detection accuracy of various driver fatigue degrees are also very similar.

## 4. Conclusions

This study uses vehicle operating data and drivers' facial video data to screen nine operating parameters of vehicles that are significantly affected by fatigue degree. A fatigue driving detection model is established using *k*-nearest neighbor algorithm, and different model parameters are set to optimize the model. The results show that compared with the fatigue driving detection model established by BP neural network, the fatigue driving detection model established by the *k*-nearest neighbor algorithm has obvious advantages in detection accuracy and stability. The principle is simple, and there are fewer parameters that are easy to calibrate accurately. When the number of training samples reach a certain value, the detection accuracy is basically maintained at 75.9%.

However, the careful analysis of detection accuracy of each type of fatigue degree in this study shows that due to driver's human factors, the significance of the vehicle operating parameters in the fatigue state and the severe fatigue state is weak, which leads to the detection results of the two fatigue states being easily confused. This may be different from a real situation, requiring further experimental observation and data analysis.

The limitations of the study also reflected in the following two aspects. The present method relies on an independent panel of experts fatigue rating; thus, the next step will be the model with no such rating, a model of a single driver through training, and then the use of the same model to predict the same driver fatigue state in the future. Secondly, the current method cannot be carried out on a real-time basis, so the next step is to extend the model to real-time detection of fatigue state by sending a fatigue state detection signal to the supervisor, so that the driver can be ordered to stop driving and rest.

**Author Contributions:** Conceptualization: J.X., S.W. and T.D.; methodology: J.X., S.W. and J.T.; software: S.W., T.D. and H.S.; validation: J.X., S.W. and T.D.; formal analysis: J.X., J.T. and H.S.; investigation: J.X., S.W. and X.M.; data curation: J.X., S.W.; writing—original draft preparation: J.X., S.W. and T.D.; writing—review and editing: J.T., H.S. and X.M.; visualization: J.X., S.W.; supervision: T.D. and J.T.; project administration: J.X.; funding acquisition: J.X. All authors have read and agreed to the published version of the manuscript.

**Funding:** This research was funded by the National Natural Science Foundation of China, grant number 51308249.

**Institutional Review Board Statement:** Not applicable.

**Informed Consent Statement:** Not applicable.

**Data Availability Statement:** The data used to support the study have not been made available because that it involves the commercial confidentiality of Tianjin SOTEREA Automotive Technology Co., Ltd. The data includes vehicle operating data and drivers' facial videos.

**Acknowledgments:** This study is supported by the China Academy of Transportation Sciences, and Beijing e.Hualu Information Technology Co., Ltd. Data used came from Tianjin SOTEREA Automotive Technology Co., Ltd.

**Conflicts of Interest:** The authors declare no conflict of interest.

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
