# Peer review of "Detection Model on Fatigue Driving Behaviors Based on the Operating Parameters of Freight Vehicles"

_applsci, doi:10.3390/app11157132_

Round 1
Reviewer 1 Report
Detection Model on Fatigue Driving Behaviors Based on the Operating Parameters of Freight Vehicles
Research on fatigue driving is the one that will always make sense because it can contribute to reducing the number of road accidents.
While fatigue driving studies using real-world vehicle performance data raise ethical questions.
This study collected a large amount of actual driving data to extract examples of data on varying degrees of fatigue.
Vehicle performance data parameters were selected on the basis of significant degrees of driver fatigue.
The k-nearest neighbor algorithm was used to establish a model of fatigue behavior detection while driving.
The results of the driver fatigue analysis based on the k-nearest neighbor algorithm may not be entirely novel, but simple and reliable to implement.
There are a few shortcomings at that article.
The statistics quoted should be a representation of the origin.
- Whether in developing or developed countries, traffic accidents caused by freight vehicles line 9
- are responsible for more than 10% of deaths of all traffic accidents. line 10
Please provide the source of this thesis.
Next.
-The expert scoring method based on facial video is the most practical method for line 129
-evaluating the fatigue degrees of drivers. This method uses a group of trained experts to line 130
-score the fatigue degrees of drivers according to their facial expressions and head posture line 131
[31]. line 132
To what extent is the assessment of the driver's condition arbitrary? Admittedly, this is not the main purpose of the presented research work.
But the arbitrariness, experts' assessments, could be confirmed by the drivers themselves (by questionnaires - suggestion).
We would then receive an additional indicator of the objectivity of assessments as the ratio of confirmed cases of expert assessment.
Next.
-The operating parameters of the vehicle directly obtained from the system mainly line 155
-include heading angle, roll angle, speed, lateral acceleration and longitudinal acceleration. line 156
-By deriving the heading angle, roll angle and speed, the heading angular velocity, roll line 157
-angular velocity and acceleration can be obtained. In this way, eight operating parameters line 158
-of the vehicle can be obtained. Taking the heading angle ? by deriving to obtain the head- line 159
-ing angular velocity ? as an example, since the vehicle operating parameter is a smooth line 160
-discrete function and the information density is large enough (the time interval is 1 s), the line 161
-increment of the independent variable when deriving the derivation is ∆? = 1, the specific line 162
-calculation formula 1 [32] - line 162
I do not fully understand the sense of enumerating these parameters. If the authors wrote earlier that:
-Therefore, line 117
-this paper collects a large number of continuous operating data of freight vehicles and line 118
-driver video data under real road traffic conditions, extracts fatigue driving detection pa- line 119
-rameters from vehicle operating data, and studies a more accurate and stable fatigue driv- lin 120
-ing detection model by using the ?-nearest neighbor algorithm. line 121
Therefore, if the driver's fatigue is assessed on the basis of video, please explain the need to define more clearly:
heading angle, roll angle, speed, lateral acceleration and longitudinal acceleration.
My guess is that the authors want to confirm the operating status of the vehicle and thus prove the driver's working time.
I suggest that this fragment be rebuilt so that it does not raise any doubts or abandon it.
The rest is presented in accordance with the art (the training set and the test bias) as follow
-For the convenience of the research, the order of 396 samples was randomly dis- line 264
-rupted. The first 79 samples (20% of the total sample number) were taken as model test line 265
-sample set including 24 alert samples, 24 fatigue samples and 31 severe fatigue samples. line 266
-The remaining samples served as training sample set. line 267
Summary of comments.
I suggest:
1. Supplementing the sources in places where statistics are indicated, or their omission.
2. Supplementing with a diagram, drawing, photo explaining the method of assessing the fatigue of drivers by three experts.
3. Explanation and clarification of the purpose for which eight vehicle parameters are calculated, since the basic data is a driver video image. Or you will skip this passage.
4. Worth refilling. Matrix of classification errors in the results.
5. It would be useful to document with a photo or map of which road area the data was obtained from.
This drawing will allow the reader to realize what kind of road and movement was documented and studied. Because the types of hazards on rural transit roads are different from those in urban areas.
6. Was there the consent of the ethics committee for the consent of the drivers to filming?
Maybe such consents are not needed? This should be documented legally.
Positive sides of the article.
1. The data set itself is unique.
2. Well-presented results confirm the knowledge of the workshop.
3. Driver fatigue is a huge research area, and at the same time very difficult. The achieved result of 75.9% on the actual data set is appreciated.
I wish the authors perseverance and good luck.
Author Response
Response to Reviewer 1 Comments
Dear Reviewers:
Thank you for your letter and for the reviewers’ comments concerning our manuscript entitled “Detection Model on Fatigue Driving Behaviors Based on the Operating State of Freight Vehicles” (ID: 8820223). Those comments are all valuable and very helpful for revising and improving our paper, as well as the important guiding significance to our researches. We have studied comments carefully and have made correction which we hope meet with approval. Revised portion are marked in yellow in the paper. The main corrections in the paper and the responds to the reviewer’s comments are as flowing:
Point 1: Supplementing the sources in places where statistics are indicated, or their omission.
Response 1: The statistics come from http://www.china.com.cn/news/txt/2017-12/19/content_42001536.htm.
Point 2: Supplementing with a diagram, drawing, photo explaining the method of assessing the fatigue of drivers by three experts.
Response 2: The indicators and process of expert scoring are shown in the Table1.
Point 3: Explanation and clarification of the purpose for which eight vehicle parameters are calculated, since the basic data is a driver video image. Or you will skip this passage.
Response 3: As the degree of fatigue deepens, the driver's ability to perceive, react, and judge will become weaker, resulting in abnormal fluctuations in vehicle control variables and state variables. In this paper, by monitoring the heading angle, roll angle, speed, lateral acceleration and longitudinal acceleration of the vehicle, the heading angle, roll angle and velocity parameter data are derived to obtain the heading angular velocity, roll angular velocity and acceleration, and the vehicle is monitored through these parameters Whether there is a deviation in the trajectory of the vehicle, whether the vehicle is driving smoothly, whether there are behaviors such as sudden braking, rapid acceleration, sharp turning, and rapid lane change.
Modify: The operating parameters of the vehicle directly obtained from the system mainly include heading angle, roll angle, speed, lateral acceleration and longitudinal acceleration. By deriving the heading angle, roll angle and speed, the heading angular velocity, roll angular velocity and acceleration can be obtained. In this way, eight operating parameters of the vehicle can be obtained. As the degree of fatigue deepens, the driver's ability to perceive, react, and judge will become weaker, resulting in abnormal fluctuations in vehicle control variables and state variables. In this paper, by monitoring the heading angle, roll angle, speed, lateral acceleration and longitudinal acceleration of the vehicle, the heading angle, roll angle and velocity parameter data are derived to obtain the heading angular velocity, roll angular velocity and acceleration, and the vehicle is monitored through these parameters Whether there is a deviation in the trajectory of the vehicle, whether the vehicle is driving smoothly, whether there are behaviors such as sudden braking, rapid acceleration, sharp turning, and rapid lane change.
Point 4:Worth refilling. Matrix of classification errors in the results.
Response 4: The final error is already in Table 5 (see Attachment).
Point 5:It would be useful to document with a photo or map of which road area the data was obtained from. This drawing will allow the reader to realize what kind of road and movement was documented and studied. Because the types of hazards on rural transit roads are different from those in urban areas.
Response 5: The driving path is an expressway in Shandong Province. Take the Yanhai Expressway as an example. The driving path is shown in the figure (see Attachment).
Point 6:Was there the consent of the ethics committee for the consent of the drivers to filming? Maybe such consents are not needed? This should be documented legally.
Response 6: The driver agreed to take the shot, because this data is collected by equipment installed by the logistics company to monitor the operating status of the driver and the vehicle at any time.

Reviewer 2 Report
Review for manuscript applsci-1299997-v1 “Detection model on fatigue driving behaviors based on the 2 operating parameters of freight vehicles” by J. Xi et al.
This manuscript reports a study of machine learning (ML) classification of real-world driving data under different degrees of fatigue. The goal was to develop a model to detect fatigued driving. The authors collected a large amount of fatigue-related driver behavior via video (e.g. eye blinks, gaze direction, yawning and breathing patterns, facial expressions, etc.) that were scored by trained experts to identify fatigue states. In addition, vehicle operating parameters were also measured (e.g. vehicle heading, velocity, acceleration, etc.); however, statistical testing showed that fatigue status significantly impacted only nine vehicle operating parameters that were then included in the modeling. Two ML models were constructed to detect fatigue driving behaviors, a k-nearest neighbor (kNN) model and a backpropagation (BP) neural network model. The study examined the effect of different model parameters (e.g. k-values, number of training/test samples, training times). The authors found that both models could achieve comparable classification accuracy (~75.9%) using different parameter values and numbers of training samples. However, the kNN model was simpler to implement and provided greater reliability under different parameter manipulations. However, whereas the models could reliably distinguish between fatigued and non-fatigued states, they had more difficulty distinguishing between moderate and sever fatigue.
I found the goal and overall approach of this study to be interesting and important, as development of a non-intrusive real-time fatigue detector for driving is of great practical value. In my assessment, the methodology is mostly satisfactory and has been reported adequately (but see comments below). That said, I do have a few questions and editorial suggestions about the present manuscript that, if addressed, could make the study’s overall readability and impact much stronger.
p. 3 – 4, Section 2.1: what was the criteria by which the expert scores were considered consistent; when all experts rated the video with the same score? Did the authors make any effort to quantify observer agreement via a relevant statistic, such as Cronbach’s alpha?
p. 4, lines 145 – 148: The authors state “Taking into account factors such as data collection errors, short-term temporary parking and other factors that do not relieve driver fatigue, the duration when operating velocity is continuously 0 km/h does not exceed 200 s, and the vehicle is regarded as in operation”. I am confused by this statement; if the operating velocity is continuously 0 km/h, doesn’t that mean the vehicle is not in operation?
p. 5, Section 2.2.2: The parameters used in the ML models were chosen on the basis of statistical tests of the effects of fatigue on these parameters. Is there any concern of statistical circularity using this method that could bias the ML algorithms towards higher accuracy?
p. 7 – 8, lines 264 – 267: The authors state” For the convenience of the research, the order of 396 samples was randomly disrupted. The first 79 samples (20% of the total sample number) were taken as model test sample set including 24 alert samples, 24 fatigue samples and 31 severe fatigue samples. The remaining samples served as training sample set”. Is the number of test samples (79 samples) the value of variable l in equation 7, i.e. “the first ? samples are extracted from the sample set ?∗” (as stated on p. 6, line 214)? Also, why was a different number of severe fatigue samples used relative to the alert and fatigue samples?
p. 8, lines 269 – 272: The authors state “In order to test the effectiveness of the fatigue driving behavior detection model based on the ?-nearest neighbor algorithm, the model (referred to as Model 1) was compared with existing fatigue driving behavior detection model based on BP neural network (referred to as Model 2)”. The authors introduce the use of the BP neural network without discussion of its structure, parameters, and properties. This information is presented later on pages 9 – 10, lines 317 – 332. This information should be included in the methods; the inclusion of a figure illustrating the basic structure of the BP network would be helpful. Also, it would be helpful to the reader to mention that the performance of the kNN classifier would be compared to a BP neural network. I suggest stating this at the end of the introduction of the paper, when the kNN classifier is first discussed. Finally, I assume BP refers to “backpropagation”; this should be explicitly stated.
p. 9, Table 4: This table lists accuracy for given a k value and number of training samples. Were the training samples chosen such that there was a roughly equal number of points associated with each different fatigue state? Or did different training samples have different proportions of points associated with each fatigue state?
p. 9, Tables 5 and 6: Are these tables based off of the detection results with the highest accuracy? This is what seems to be stated in the text on lines 300 – 301. If so, it would be helpful to state this either in the table titles of a caption/note for each table.
p. 13, Tables 8 and 9: Again, are these tables based off of the detection results with the highest accuracy? This is what seems to be stated in the text on lines 378 – 379. If so, it would be helpful to state this either in the table titles of a caption/note for each table.
p. 13 – 14, Section 4. Conclusions: The authors should include a discussion of the limitations of their study. There are two major limitations that I can identify (there are likely more that I haven’t identified). First, the current method relies on the fatigue ratings of an independent group of experts. How can this model be extended to the case where such ratings are unavailable? Could one train a model using this method for a single driver and then use that same model to predict fatigue states of that same driver in the future? Second, the current method does not seem to be able to be carried out in a real-time basis where it would be most useful (perhaps by signaling fatigue state detection to a supervisor who could order a driver to stop driving and rest). How might the model be extended to detect fatigue states on a real-time basis?
Author Response
Response to Reviewer 2 Comments
Dear Reviewers:
Thank you for your comments concerning our manuscript entitled “Detection Model on Fatigue Driving Behaviors Based on the Operating Parameters of Freight Vehicles” (Manuscript ID: applsci-1299997). Those comments are all valuable and very helpful for revising and improving our paper, as well as the important guiding significance to our researches. We have studied comments carefully and have made correction which we hope meet with approval. Revised portion are marked up using the “Track Changes” function. The main corrections in the paper and the responds to the reviewer’s comments are as flowing:
Point 1: p. 3 – 4, Section 2.1: what was the criteria by which the expert scores were considered consistent; when all experts rated the video with the same score? Did the authors make any effort to quantify observer agreement via a relevant statistic, such as Cronbach’s alpha?
Response 1: The result of expert concordant evaluation is that three experts give the same score to the same video.Driver fatigue state is divided into awake, tired, very tired, and the corresponding scores respectively 1, 2, 3, 3 expert a character description of evaluation standards on the same segment of the video driver fatigue rating, the fatigue status of if three expert evaluation is consistent, namely the consistency of expert evaluation results as the sample fatigue level.
As the experts invited this time are all experienced related personnel of the enterprise, who have a profound understanding of the fatigue driving state; In addition, the tested drivers are experienced drivers who know whether they are tired when driving, so the consistency test is not carried out this time.
Modify: If the three experts score the same, that is, the experts' consensus evaluation result is taken as the fatigue level of the sample When the scoring results are inconsistent, the three experts will conduct a collegiate discussion. If they are consistent after the negotiation, the result of will be used as the fatigue degree of the sample. If there is still a disagreement after the negotiation, the sample will be discarded.
Point 2: p. 4, lines 145 – 148: The authors state “Taking into account factors such as data collection errors, short-term temporary parking and other factors that do not relieve driver fatigue, the duration when operating velocity is continuously 0 km/h does not exceed 200 s, and the vehicle is regarded as in operation”. I am confused by this statement; if the operating velocity is continuously 0 km/h, doesn’t that mean the vehicle is not in operation?
Response 2: In the text, temporary parking is equivalent to no parking.
Modify: the duration when operating velocity is continuously 0 km/h does not exceed 200 s, and the vehicle is regarded as in a temporary stop state. This paper considers that the vehicle is still in operation.
Point 3: p. 5, Section 2.2.2: The parameters used in the ML models were chosen on the basis of statistical tests of the effects of fatigue on these parameters. Is there any concern of statistical circularity using this method that could bias the ML algorithms towards higher accuracy?
Response 3: Through Friedman test and the Bonferroni adjusted multiple comparison test analysis method, this paper has extracted significant parameters, so there is no ML algorithm test.
Point 4: p. 7 – 8, lines 264 – 267: The authors state” For the convenience of the research, the order of 396 samples was randomly disrupted. The first 79 samples (20% of the total sample number) were taken as model test sample set including 24 alert samples, 24 fatigue samples and 31 severe fatigue samples. The remaining samples served as training sample set”. Is the number of test samples (79 samples) the value of variable l in equation 7, i.e. “the first ? samples are extracted from the sample set ?∗” (as stated on p. 6, line 214)? Also, why was a different number of severe fatigue samples used relative to the alert and fatigue samples?
Response 4: The sample set X* contains 396 samples, and the first 79 samples are taken as the test sample set after shuffling the order, that is, the value of variable L in Formula 7 is 79.
The number of training samples in each fatigue state is approximately equal. There are 396 samples in total, and the number of samples in each fatigue state is 132. After the model test samples selected under the corresponding fatigue state are removed, the number of training samples in awake and fatigue state is 108, and the number of training samples in severe fatigue state is 101.
Modify:For the convenience of the research, the order of 396 samples in the sample set X* was randomly disrupted. The first 79 samples (20% of the total sample number) were taken as model test sample set,that is, the value of variable L in Formula 7 is 79. in-cludingThere were 24 alert samples, 24 fatigue samples and 31 severe fatigue samples. The remaining samples served as training sample set.
Point 5: p. 8, lines 269 – 272: The authors state “In order to test the effectiveness of the fatigue driving behavior detection model based on the ?-nearest neighbor algorithm, the model (referred to as Model 1) was compared with existing fatigue driving behavior detection model based on BP neural network (referred to as Model 2)”. The authors introduce the use of the BP neural network without discussion of its structure, parameters, and properties. This information is presented later on pages 9 – 10, lines 317 – 332. This information should be included in the methods; the inclusion of a figure illustrating the basic structure of the BP network would be helpful. Also, it would be helpful to the reader to mention that the performance of the kNN classifier would be compared to a BP neural network. I suggest stating this at the end of the introduction of the paper, when the kNN classifier is first discussed. Finally, I assume BP refers to “backpropagation”; this should be explicitly stated.
Response 5: It has been revised in the article.
Modify: The BP neural network model can also effectively detect the fatigue driving behavior. Therefore, in order to test the effectiveness, the fatigue driving behavior detection model based on the K-nearest neighbor algorithm can be compared with the existing fatigue driving behavior detection model based on the BP neural network to test the effectiveness of the proposed method.
The program is developed with Matlab to realize the calculation of the fatigue driving detection model based on the BP neural network (see Figure 2).
Point 6: p. 9, Table 4: This table lists accuracy for given a k value and number of training samples. Were the training samples chosen such that there was a roughly equal number of points associated with each different fatigue state? Or did different training samples have different proportions of points associated with each fatigue state?
Response 6: Figure 4 shows the number of training sets corresponding to the maximum detection accuracy under different K values. When k value is 7, that is, the number of training sample sets is 284-306, the maximum detection accuracy of fatigue driving reaches 75.9%. Therefore, the model value should be 7. When the ratio of detection set to training sample is between 2:7 And 2:8, the requirement for training sample is low, the accuracy is high and the stability is strong.
Point 7: p. 9, Tables 5 and 6: Are these tables based off of the detection results with the highest accuracy? This is what seems to be stated in the text on lines 300 – 301. If so, it would be helpful to state this either in the table titles of a caption/note for each table.
Response 7: Table 5 and Table 6 are the test results when the test accuracy reaches the maximum,It has been supplemented in the article.
Point 8: p. 13, Tables 8 and 9: Again, are these tables based off of the detection results with the highest accuracy? This is what seems to be stated in the text on lines 378 – 379. If so, it would be helpful to state this either in the table titles of a caption/note for each table.
Response 8: Table 8 and Table 9 are the test results when the test accuracy reaches the maximum,It has been supplemented in the article.
Point 9: p. 13 – 14, Section 4. Conclusions: The authors should include a discussion of the limitations of their study. There are two major limitations that I can identify (there are likely more that I haven’t identified). First, the current method relies on the fatigue ratings of an independent group of experts. How can this model be extended to the case where such ratings are unavailable? Could one train a model using this method for a single driver and then use that same model to predict fatigue states of that same driver in the future? Second, the current method does not seem to be able to be carried out in a real-time basis where it would be most useful (perhaps by signaling fatigue state detection to a supervisor who could order a driver to stop driving and rest). How might the model be extended to detect fatigue states on a real-time basis?
Response 9: It has been supplemented in the article.
Modify: The limitations of the study also reflected in the following two aspects, the present method relies on an independent panel of experts fatigue rating, the next step will be the model to no such rating, a model of a single driver through training, and then use the same model to predict the same driver fatigue state of the future; Secondly, the current method can not be carried out on a real-time basis, the next step is to extend the model to real-time detection of fatigue state, by sending a fatigue state detection signal to the super-visor, so that the driver can be ordered to stop driving and rest.

Round 2
Reviewer 2 Report
The authors have addressed my concerns.